# Quantitative Analysis of a Pilot Transwell Barrier Model with Automated Sampling and Mathematical Modeling

**DOI:** 10.3390/pharmaceutics15112646

**Published:** 2023-11-20

**Authors:** Júlia Tárnoki-Zách, Szilvia Bősze, András Czirók

**Affiliations:** 1Department of Biological Physics, Eötvös University, 1053 Budapest, Hungary; julia.tarnoki-zach@ttk.elte.hu; 2National Center for Public Health and Pharmacy, 1437 Budapest, Hungary; szilvia.bosze@ttk.elte.hu; 3HUN-REN-ELTE Research Group of Peptide Chemistry, Hungarian Research Network, Eötvös Loránd University, 1052 Budapest, Hungary

**Keywords:** in vitro barrier model, transport kinetics, mathematical transport model, automated sample collection, fluidics, diffusive permeability

## Abstract

In the preclinical phase of drug development, it is necessary to determine how the active compound can pass through the biological barriers surrounding the target tissue. In vitro barrier models provide a reliable, low-cost, high-throughput solution for screening substances early in the drug candidate development process, thus reducing more complex and costly animal studies. In this pilot study, the transport properties of TB501, an antimycobacterial drug candidate, were characterized using an in vitro barrier model of VERO E6 kidney cells. The compound was delivered into the apical chamber of the transwell insert, and its concentration passing through the barrier layer was measured through the automated sampling of the basolateral compartment, where media were replaced every 30 min for 6 h, and the collected samples were stored for further spectroscopic analysis. The kinetics of TB501 concentration obtained from VERO E6 transwell cultures and transwell membranes saturated with serum proteins reveal the extent to which the cell layer functions as a diffusion barrier. The large number of samples collected allows us to fit a detailed mathematical model of the passive diffusive currents to the measured concentration profiles. This approach enables the determination of the diffusive permeability, the diffusivity of the compound in the cell layer, the affinity of the compound binding to the cell membrane as well as the rate by which the cells metabolize the compound. The proposed approach goes beyond the determination of the permeability coefficient and offers a more detailed pharmacokinetic characterization of the transwell barrier model. We expect the presented method to be fruitful in evaluating other compounds with different chemical features on simple in vitro barrier models. The proposed mathematical model can also be extended to include various forms of active transport.

## 1. Introduction

In vitro barrier models are helpful tools in preclinical research to evaluate the uptake and transport properties of drug candidates [1,2,3,4] or tissue-specific targeting compounds [5]. Currently, the two main experimental approaches to analyze trans-barrier transport utilize either cell cultures and transwell diffusion chambers or excised tissue pieces placed into much larger devices like the Franz and Ussing chambers. In the Franz chamber, the specimen (usually a skin sample or skin equivalent) separates two liquid compartments [6,7,8,9,10,11]. The receptor chamber is well mixed and temperature-controlled but usually not kept in tissue culture incubators. The size of the standard Franz chamber requires large and mechanically robust samples that are difficult to achieve with cell culture-based approaches [12,13]. In order to overcome the limitations mentioned above, various microfluidic Franz chambers have been proposed that allow both automated sample collection and the utilization of substantially smaller samples [14,15,16,17,18,19,20]. The requirement of a continuous microfluidic flow, however, must be provided with a non-standard tissue culture infrastructure and also makes high throughput parallel measurements challenging to achieve.

The other often used approach to study transepithelial transport utilizes cell layers cultured on the porous membrane of transwell inserts [1,2,3]. Transwell barrier models are handled using standard cell culture tools, allowing parallelization for high throughput studies. Test compounds are introduced into the apical compartment and transported through the barrier into the basolateral compartment of the transwell device (Figure 1a). In order to characterize transport kinetics, the amount of compound within the basolateral compartment is measured either directly, using spectroscopy methods (UV-VIS, fluorescence or mass spectroscopy), or indirectly using biological assays [5,21,22,23]. The latter approach often involves a second cell layer of detector cells, which can take up the experimental compound from the basolateral compartment. At the conclusion of the experiments, the detector cells are harvested and their uptake of the compound is evaluated using flow cytometry [5,22,23]. As a specific transwell culture is evaluated at a single time point, kinetic studies require parallel cultures.

In this paper, as a pilot study, we employ an automated millifluidic system to characterize the transport through a VERO E6 kidney model barrier culture. The integrity of the barrier was monitored through daily transepithelial electrical resistance (TEER) measurements. As a pharmacological test compound, we used TB501 [24], a promising antimycobacterial agent [25,26]. TB501 was identified by a FRIGATE software docking screen [24] involving the high-resolution crystal structure of *M. tubercolosis* dUTPase as the target [27,28,29,30] and over 13 million small molecules from the ZINC database as drug candidates [31].

Advances in millifluidics [32,33] allow the automatization of standard cell culture procedures, including periodic sample collection under sterile conditions [34,35,36]. This approach can obtain enough samples from regular transwell barrier cultures to determine their time-resolved transport kinetics comparable to the data obtainable from microfluidic Franz chambers. We also develop a mathematical model of the sample collection process that can be used to estimate and compensate for adsorption artifacts and thus provide a more accurate estimate of the diffusive transport. As we demonstrate, the time-resolved concentration data allow us to determine parameters of the cellular barrier, such as the diffusive permeability of the test compound, its affinity to the cell membrane, its diffusivity through the cellular barrier, and the rate of its metabolic elimination.

## 2. Materials and Methods

### 2.1. Reagents, Buffers, and Media

For the in vitro assays, Dulbecco’s Modified Eagle’s Medium (DMEM) without phenol red, phosphate-buffered saline (PBS), and L-glutamine were obtained from Lonza (Basel, Switzerland). Pyruvate and trypsin were obtained from Sigma-Aldrich (St. Louis, MO, USA). Non-essential amino acids, fetal bovine serum (FBS), and Penicillin/Streptomycin (10,000 units penicillin and 10 mg streptomycin/mL) were purchased from Gibco (Thermo Fisher Scientific, Waltham, MA, USA).

For maintaining the cell cultures in transwell inserts, DMEM without phenol red was supplemented with 10% FBS, 2 mM L-glutamine, 100 µg/mL Penicillin/Streptomycin, 1 mM Pyruvate, and 1% non-essential amino acids—hereafter referred to as complete medium (CM). The same medium without serum and non-essential amino acids is referred to as the incomplete medium (ICM). Both CM and ICM were set to pH 7.4.

HPMI buffer [37] at pH 7.4 was prepared in our laboratory using components (9 mM glucose, 10 mM NaHCO${}_{3}$, 119 mM NaCl, 9 mM HEPES (N-(2-hydroxyethyl)piperazine-N′-(2-ethanesulfonic acid)), 5 mM KCl, 0.85 mM MgCl${}_{2}$, 53 μM CaCl${}_{2}$, 5 mM Na${}_{2}$HPO${}_{4}$× 2 H${}_{2}$O) obtained from Sigma-Aldrich.

TB501 (UIPAC name: 6-hydroxy-7-[4-(2-hydroxyethyl)piperazin-1-yl]methyl-2-[(2E)-3-(2-methoxy phenyl)prop-2-en-1-ylidene]-2,3-dihydro-1-benzofuran-3-one) is an apolar compound (logP=1.523) with good intrinsic solvability (−3.435) and a molecular weight of $M_{w}=436.51$ Da. The compound was purchased from SONEAS Research (formerly Ubichem Research, Budapest, Hungary.) We prepared a diluted solution for the experiments by mixing a 20 mM TB501 DMSO stock solution with ICM or HPMI buffer.

### 2.2. Transwell Culture

Green monkey (*Chlorocebus sabaeus*) kidney epithelium VERO E6 cells [38,39,40,41] were kindly provided by Bernadett Pályi and Zoltán Kis (National Public Health Center, Budapest, Hungary) based on the antiviral project EFOP-1.8.0-VEKOP-17-2017-00001, and were obtained from the European Collection of Authenticated Cell Cultures (ECACC 85020206). Cultures were maintained in CM and kept in a humidified, 5% CO${}_{2}$ incubator at 37 °C.

For barrier transport experiments, the polycarbonate transwell inserts (0.6 cm${}^{2}$ area, 0.4 μm pore size, Merck Millipore, Burlington, MA, USA) were equilibrated for at least 2 h in 12-well plates (Sarstedt, Nümbrecht, Germany) with CM, then into each transwell insert 75,000 cells were seeded in 500 μL CM. At the same time, 1800 μL CM was added to the basolateral compartment. Cultures were grown for 5–7 days under standard culture conditions [5,22,23]. Before cell-free transport measurements, transwell inserts were saturated with serum proteins by incubation in CM for 5–7 days. Corresponding unsaturated control transwell inserts were incubated in ICM.

Immediately before barrier transport measurements, transwell cell cultures and saturated cell-free transwell inserts were washed twice with PBS, and then the apical and basolateral chambers were filled with ICM. In some cell-free transport measurements, we used HPMI buffer. The experimental compound was introduced into the apical chamber. Media from the basolateral compartment were collected and replaced with fresh medium every 30 min. Collected samples were stored for further spectroscopic analysis. After the conclusion of the experiments, we recorded the VIS absorbance spectra of the samples, the culture media, and the solution loaded into the apical chamber. Cellular barrier models were always measured with ICM and kept in a 5% CO${}_{2}$ tissue culture incubator. Some cell-free calibration measurements were carried out with HPMI buffer at normal atmospheric CO${}_{2}$ level.

### 2.3. Transepithelial Electrical Resistance

Millicell ERS-2 voltohmmeter (Merck, Darmstadt, Germany) equipped with MERSSTX01 chopstick electrodes (Merck, Darmstadt, Germany) was used to measure transepithelial electrical resistance (TEER) of transwell inserts before barrier transport experiments. TEER was calculated as the difference in the resistance of cell-containing and cell-free transwell inserts, normalized with the cell culturing area.

### 2.4. Spectroscopy

VIS absorption spectra were recorded with a Shimadzu UV-2101PC double beam spectrophotometer (Shimadzu, Kyoto, Japan) in the 370–750 nm spectral range at a spectral resolution of 2 nm, using distilled water as a reference. Samples were equilibrated for at least 30 min at normal atmospheric CO${}_{2}$ conditions and loaded into 1 mL polystyrene cuvettes with 1 cm optical path length (Sigma-Aldrich, St Louis, MO, USA). ICM or HPMI were used in spectroscopy analyses to avoid absorption from phenol red and serum proteins. Both DMEM and HPMI culture media, as well as the TB501 compound, exhibit absorption peaks below 550 nm (see Figure 1). Thus, a linear fit in the 550–750 nm spectral range was subtracted from the recorded spectra for baseline correction. Medium-referenced spectra were obtained as the difference between the baseline corrected spectra of the sample and the corresponding medium.

To determine the concentration of TB501 from medium-referenced spectra, we recorded the absorption spectra of a sequence of solutions within the concentration range of 13.3 nM and 10 μM (Figure 1). For DMEM-based solutions, we used absorbance values $A_{418}$ measured at the absorption peak, 418 nm. As HPMI exhibits absorbance near this spectral line, the TB501 content of HPMI-based solutions was characterized with the off-peak absorbance $A_{450}$, measured at 450 nm. The obtained calibration data can be fitted with a linear function for absorbance values above 0.03 and with a power law below (Figure 1). Accordingly, we used either the linear or the power law fit to calculate concentrations depending on the magnitude of absorbance. In particular, we extracted TB501 concentration from DMEM-based solutions as
(1)$$\left[TB501\right]=\min\left(25.5\cdot A_{418},{\left(24\cdot A_{418}\right)}^{1.0526}\right)\;,$$
while from HPMI-based solutions as
(2)$$\left[TB501\right]=\min\left(45\cdot A_{450},{\left(55\cdot A_{450}\right)}^{1.538}\right)\;,$$
where the min function selects the smaller one of its two arguments and thus provides a convenient formula to use either the linear or the power law fit depending on the absorbance magnitude.

### 2.5. Automated Sample Handling

We used an automated sampling system (Millitransflow Fluidic System, BioPhys-Concepts, Budapest, Hungary, biophys-concepts.com, accessed on 11 November 2023) for the measurements. The sampling process utilized two separate tubing systems: one delivered the fresh medium while the other handled the sample removed from the culture well. The dosing unit used two syringe pumps and a tube multiplexer to measure and inject 1.8 mL of cooled media into the tube connected to the incubator unit. A peristaltic pump removed the TB501-containing medium from the culture well. Samples were collected in 2 mL Eppendorf tubes in the cooled fraction collector of the sampling system. Within the culture incubator, the fresh medium was kept in a heat exchanger for 30 min before its injection into the culture well. The time delay between the complete removal of the basolateral medium and the injection of fresh medium was less than 10 s.

Between measurements, the tubing network was cleaned and disinfected overnight. First, all fluid handling volumes were filled for 20 min with 0.1% sodium hypochlorite. After the removal of the hypochlorite, the tubing network was rinsed six times with sterile distilled water.

### 2.6. Computational Methods

Python codes were used to process UV-VIS spectra, fit model parameters, and analyze sample sequences. Computational code and data are available at the Open Science Framework (osf.io/sc3wq, accessed on 11 November 2023). Parameter fitting was performed using the scipy.optimize.minimize function, which uses the quasi-Newton method of Broyden, Fletcher, Goldfarb, and Shanno (BFGS) [42]. The diffusion model was numerically integrated using the explicit finite difference (Euler) method.

#### 2.6.1. Adsorption Effects during Sample Collection

Let us denote by $S_{i}$, $x_{i}$, and $y_{i}$ the amount of experimental compound measured in sample *i*, left behind as droplets and adsorbed to surfaces when sample *i* is collected, respectively. If $I_{i}$ denotes the amount of test compound introduced into the culture volume during the time interval between taking the consecutive samples i−1 and *i*, then the total amount of test compound collected during this time interval is
(3)$$T_{i}=I_{i}+x_{i-1}+k_{off}\cdot y_{i-1}$$
where $k_{off}$ is a parameter proportional to the desorption rate of the immobilized compound, the duration of contact, and the contact area between the sample and the adsorbing surface, factors that remain unchanged during sample collection. Due to adsorption and surface tension, sample *i* will contain only
(4)$$S_{i}=T_{i}\cdot \left(1-k_{on}\right)\cdot \left(1-q\right)$$
amount of test compound, where $k_{on}$ characterizes the adsorption process, and *q* is the volume fraction of the liquid the sampling procedure leaves behind in droplets. Thus, according to the model visualized in Figure 2a, after the collection of sample *i*, the amount of adsorbed compound is
(5)$$y_{i}=k_{on}\cdot T_{i}+\left(1-k_{off}\right)\cdot y_{i-1},$$
while the amount of test compound remaining in droplets is
(6)$$x_{i}=\left(1-k_{on}\right)\cdot q\cdot T_{i}.$$

**Figure 2 pharmaceutics-15-02646-f002:**
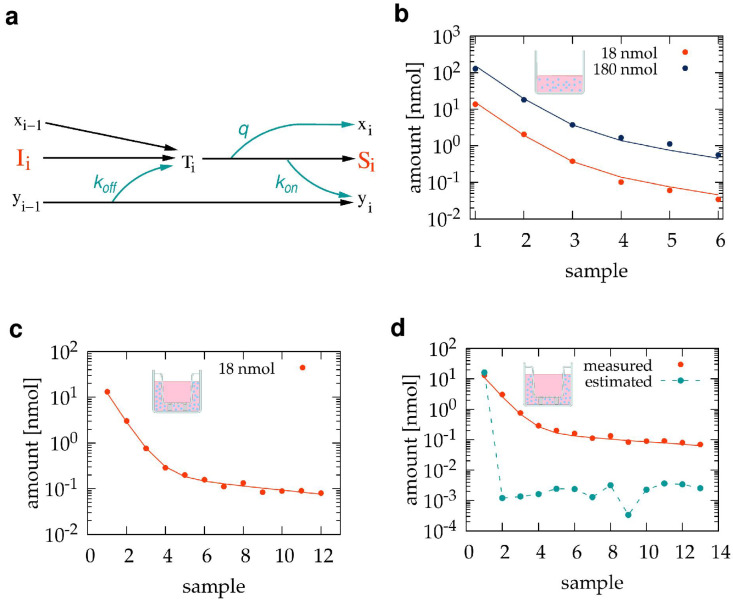
Mathematical model of the sampling procedure. (**a**) Graphical representation of Equations (Equation 3)–(Equation 6) that predict $S_{i}$, the test compound content of sample *i* given the parameters $k_{on}$, $k_{off}$, and *q*. (**b**–**d**) TB501 content of sample sequences obtained from calibration experiments and their best-fitting model simulations. A well of a 12-well dish was initially loaded with 1.8 mL HPMI cell culture medium containing either 10 (red) or 100 (blue) μM TB501 compound in the absence (**b**) or presence (**c**,**d**) of a transwell insert. (**b**,**c**) Data points represent spectrophotometrically determined TB501 content within the collected samples. Solid lines indicate best fitting model simulations, generated using the initial amount of compound $I_{1}$ and parameter values given in Table 1. (**d**) The fitted mathematical model can estimate $I=\{I_{1},I_{2},...,I_{N}\}$ (cyan), the amounts of TB501 that were added to the culture chamber during the individual sampling periods. The fitting procedure selects the sequence *I* which best reproduces (solid line) the experimentally observed TB501 content profile (red).

**Table 1 pharmaceutics-15-02646-t001:** Fitted parameters characterizing the sample collection process, both in the absence and presence of a transwell insert.

Parameter	Value	
	**Without Transwell**	**With Transwell**
$k_{on}$	0.034	0.088
$k_{off}$	0.400	0.110
*q*	0.110	0.240

The parameters $k_{on}$, $k_{off}$, and *q* thus characterize the sampling process. These parameters can be determined through calibration measurements that initially load the culture well with a known amount of test compound, $I_{1}$, and evaluate its dilution in the sample sequence (Figure 2b,c). Thus, $I_{i}=0$ for i>1 and Equations (Equation 3)–(Equation 6) are used to predict the test compound content of samples $S=\{S_{1},S_{2},...,S_{N}\}$ for a given set of parameters $k_{on}$, $k_{off}$, and *q*. The fitting algorithm selects the parameter set that minimizes the difference $\Delta (S,\tilde{S})$ between the test compound contents of the measured ($\tilde{S}$) and simulated (*S*) sample sequences. The comparison is performed on a logarithmic scale, as the compound is diluted by several orders of magnitude:(7)$$\Delta^{2}\left(S,\tilde{S}\right)=\sum_{i=1}^{N}{\left(\log S_{i}-\log{\tilde{S}}_{i}\right)}^{2}.$$

The fitting process uses the BFGS algorithm [42], started from a user-provided initial estimate.

Once parameters $k_{on}$, $k_{off}$, and *q* are determined, the mathematical model (Figure 2a) can also be used to estimate the amounts of test compound introduced in each sampling period $I=\{I_{1},I_{2},...,I_{N}\}$ from the measured compound content of the samples $\tilde{S}$. This goal is accomplished by using Equations (Equation 3)–(Equation 6) to simulate samples for various hypothetical input sequences *I*. Again, comparing the TB501 content of the simulated (*S*) and measured ($\tilde{S}$) sample sequences allows us to identify the input sequence *I* for which $\Delta (S,\tilde{S})$ is minimal.

#### 2.6.2. Adsorption Model Distinguishing Sub-Compartments

If superscripts *b* and *ℓ* denote variables for the basal and lateral sub-compartments (Figure 1a), respectively, then the equations corresponding to Figure 3a are
(8)$$T_{i}^{b}=I_{i}^{b}+x_{i-1}^{b}+k_{off}^{b}\cdot y_{i-1}^{b}$$
(9)$$y_{i}^{b}=k_{on}^{b}\cdot T_{i-1}^{b}+\left(1-k_{off}^{b}\right)\cdot y_{i-1}^{b}$$
(10)$$x_{i}^{b}=\left(1-k_{on}^{b}\right)\cdot q^{b}\cdot T_{i}^{b}$$
(11)$$T_{i}^{\ell }=I_{i}^{\ell }+T_{i}^{b}\cdot \left(1-k_{on}^{b}\right)\cdot \left(1-q^{b}\right)+x_{i-1}^{\ell }+k_{off}^{\ell }\cdot y_{i-1}^{\ell }$$
(12)$$y_{i}^{\ell }=k_{on}^{\ell }\cdot T_{i}^{\ell }+\left(1-k_{off}^{\ell }\right)\cdot y_{i-1}^{\ell }$$
(13)$$x_{i}^{\ell }=\left(1-k_{on}^{\ell }\right)\cdot q^{\ell }\cdot T_{i-1}^{\ell }$$
(14)$$S_{i}^{\ell }=T_{i}^{\ell }\cdot \left(1-k_{on}^{\ell }\right)\cdot \left(1-q^{\ell }\right).$$

Thus, the sub-compartment model characterizes the sample correction process with six parameters: $k_{on}^{\ell }$, $k_{off}^{\ell }$, $q^{\ell }$, $k_{on}^{b}$, $k_{off}^{b}$, and $q^{b}$. These parameters can be determined by parallel-fitting two kinds of calibration measurements. The first is analogous to the one considered in the previous section; we initially load the culture well with a known amount of experimental compound ($B_{1}$), insert the transwell device with no test compound in the apical compartment, and determine how the compound is diluted in the sample sequence ${\tilde{S}}^{B}$ (Figure 3b). Thus, $I_{1}^{b}=rB_{1}$ and $I_{1}^{\ell }=\left(1-r\right)B_{1}$ where *r* is the volume fraction of the medium under the transwell insert, and $I_{i}^{b}=I_{i}^{\ell }=0$ for i>1. As in the previous approach, Equations (Equation 8)–(Equation 14) are used to predict the test compound content of the samples *S* for various sets of parameters, and the set is chosen that minimizes $\Delta ({\tilde{S}}^{B},S)$. In the second kind of calibration experiment (Figure 3c), the apical compartment is loaded with a known amount of test compound ($A_{1}$) and then the compound content ${\tilde{S}}^{A}$ of the corresponding sample sequence is measured. Again, using the parameters determined from ${\tilde{S}}^{B}$ and Equations (Equation 8)–(Equation 14), we estimate the transmembrane diffusive currents $I_{i}^{b}$ by minimizing $\Delta ({\tilde{S}}^{A},S)$. We also require the estimated transmembrane diffusive currents asymptotically add up to $A_{1}$. Thus, we demand that the model parameters also minimize
(15)$$\Delta_{a}^{2}\left(I^{b},A_{1}\right)={\left(\lim_{N\to \infty }\sum_{i=1}^{N}I_{i}^{b}-A_{1}\right)}^{2}.$$

**Figure 3 pharmaceutics-15-02646-f003:**
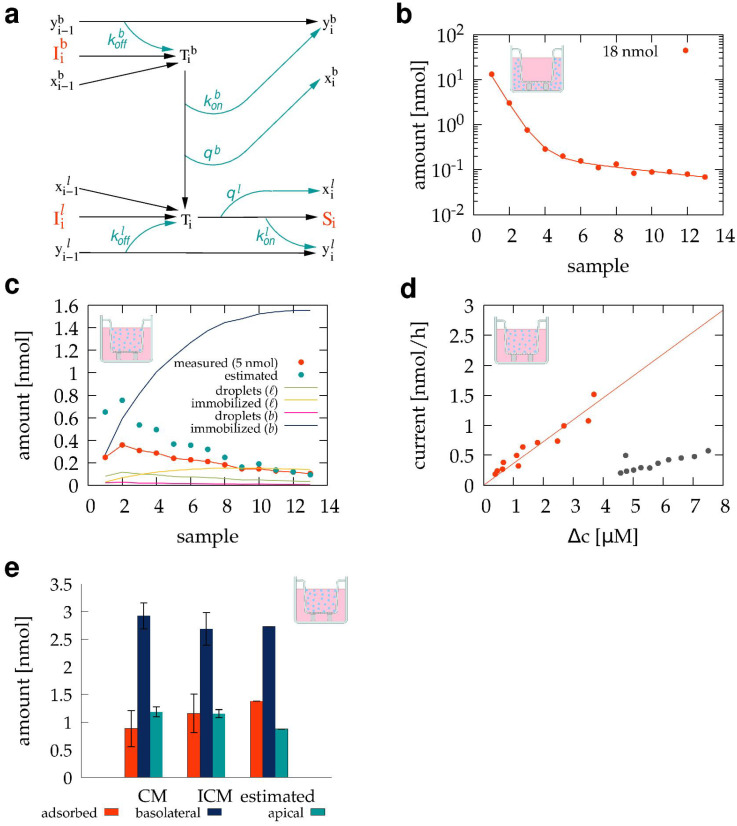
Mathematical model of the sampling procedure distinguishing basal and lateral volume compartments. (**a**) Graphical representation of Equations (Equation 8)–(Equation 14) that predict the amount of TB501 in sample $S_{i}$ given the six parameters that characterize the process. (**b**,**c**) TB501 content of sample sequences obtained from calibration experiments and their best-fitting model simulations. A transwell insert-containing well of a 12-well dish was initially loaded with 10 μM TB501 solution, either in the basolateral (**b**) or the apical (**c**) volume compartment. Data points indicate spectrophotometrically determined TB501 content within the collected samples. Solid lines represent best-fitted model predictions, generated with the parameter values given in Table 2, for the TB501 content of the samples (red), TB501 adsorbed to the transwell membrane (dark blue) or the culture and fluid handling system (yellow), or remaining in droplets either under the transwell membrane (pink) or in the rest of the system (green). The model was also used to obtain a best-fit estimate for the amount diffused through the transwell membrane (light blue). (**d**) Diffusion permeability of the transwell membrane, where the transmembrane current and concentration difference values are predicted using the sub-compartment model (red), or by using uncorrected data (gray). (**e**) Amounts of TB501 within the apical (light blue) and basolateral (dark blue) liquid compartments after a 6 h long exposure, as well as the amount adsorbed to the insert (red). Bar clusters compare values obtained from membranes that were pre-conditioned either with CM or ICM and model predictions.

**Table 2 pharmaceutics-15-02646-t002:** Fitted parameters characterizing the sample collection process, distinguishing the basal and lateral volume fractions in the presence of a transwell insert. Error estimates are standard deviations from triplicate measurements.

Basal Sub-Volume		Lateral Sub-Volume	
**Parameter**	**Value**	**Parameter**	**Value**
$k_{on}^{b}$	0.410±0.010	$k_{on}^{\ell }$	0.081±0.001
$k_{off}^{b}$	0.037±0.005	$k_{off}^{\ell }$	0.115±0.001
$q^{b}$	0.053±0.003	$q^{\ell }$	0.246±0.001

We employed the following procedure for the parallel minimization of two goal functions. For a given value of $k_{on}^{b}$, we determined the rest of the parameters by minimizing $\Delta ({\tilde{S}}^{B},S)$. Then, we estimated $I^{b}$ using the complete set of model parameters. We finally determined $\Delta_{a}^{2}\left(I^{b},A_{1}\right)$ by fitting an exponential function to the $I^{b}$ sequence to extrapolate $\Delta_{a}^{2}\left(I^{b},A_{1}\right)$. Repeating this procedure for various values of $k_{on}^{b}$, we select the value for which $\Delta_{a}^{2}\left(I^{b},A_{1}\right)$ is minimal.

The sub-compartment model can also be used to estimate the concentration difference between the apical and basal surfaces of the membrane. The amount of compound in the apical volume after the removal of sample *i* is
(16)$$A_{i}=A_{1}-\sum_{k=1}^{i}I_{k}^{b}.$$

For sample *i*, the average amount of compound in the basal compartment is $T_{i}^{b}/2$, assuming a linear accumulation from a compound-free state at the conclusion of the previous sample collection. Thus, for sample *i*, the characteristic concentration difference between the apical and basal barrier surfaces is
(17)$$\Delta c_{i}=\frac{A_{i}}{V_{a}}-\frac{T_{i}^{b}}{2V_{b}}$$
where $V_{a}=500$ μL and $V_{b}=110$ μL are the apical and basal volumes of the transwell insert, respectively.

## 3. Results

### 3.1. Pilot Transwell Transport Measurement

In this pilot study, as a test compound, we used TB501 [24,25,26,43], an antimycobacterial agent, and VERO E6 cells as a barrier model. Dissolved TB501 can be detected using absorption spectrophotometry utilizing an absorption peak at 418 nm both in ICM and HPMI (Figure 1b,c). In our barrier transport experiments, TB501 was introduced into the apical compartment of a transwell device, while the medium in the basolateral compartment was replaced every 30 min (Figure 1a).

### 3.2. Estimating and Correcting Sampling Artifacts

The TB501 content of the sample sequence, generated during the automated sampling procedure, can differ from the amount diffusing through the barrier from the apical compartment. The TB501 binding to and release from the surfaces of fluid handling tubes and the culture environment are essential sources of this discrepancy. Apolar compounds, especially, are expected to bind plastic surfaces with high affinity, i.e., exhibiting high adsorption and low desorption rates. Furthermore, during the transport of liquid samples, surface tension effects can cause droplets to remain within the culture well and inside the tubing. While the amount of TB501 within the droplets is missing from the collected sample, the droplets will be merged with the following sample. This contamination of subsequent samples can be estimated and mitigated using the mathematical model described in the Section 2 and visualized in Figure 2a.

The parameters that best characterize the sampling process in the absence or presence of a transwell insert are given in Table 1. As Figure 2b demonstrates, the same set of parameters can fit experiments where the amount of the initially introduced compound was changed by a factor of ten. This finding supports the assumption that the availability of binding sites in the Langmuir adsorption kinetic model [44] is not a limiting factor in our experiments. The presence of a transwell insert increases both the liquid volume that remains in droplets during sample collection and the amount of TB501 immobilized on surfaces (Table 1). The increase in droplet volume is due to surface tension keeping a liquid film between the bottom of the insert membrane and the culture dish surface. The increased adsorption of the compound likely reflects that the porous membrane has a large surface area in which apolar compounds can bind with high affinity.

The above approach is validated by correctly predicting that, in the TB501 rinsing experiment shown in Figure 2c, four orders of magnitude more TB501 compound was added to the first sample than to any of the subsequent samples (Figure 2d). Moreover, the predicted initial amount of TB501, $I_{1}=16$ nmol, is 89% of the correct value ${\tilde{I}}_{1}=18$ nmol, while the measured amount in the first sample, ${\tilde{S}}_{1}=7.3$ nmol, is only 40% of ${\tilde{I}}_{1}$. This approach can thus mitigate artifacts of the sampling process to a large extent.

### 3.3. The Sub-Compartment Mathematical Model of Transwell Transport Measurements

The 1 mm thin basal volume under the transwell membrane (Figure 1a) restricts the mixing of the basolateral compartment. According to our observations, compounds diffusing from the apical compartment first accumulate under the basal membrane surface and only mix with the rest of the basolateral compartment when the sampling process introduces sustained fluid flows in the culture chamber. As diffusion across the transwell membrane is driven by the concentration difference between the membrane surfaces, this diffusive current is sensitive to the accumulation of TB501 in the basal sub-compartment. Thus, we augmented the sampling model (Figure 2a) to distinguish the basal and lateral sub-compartments.

The sub-compartment model, described in the Section 2 and represented in Figure 3a, includes specific adsorption and desorption parameters for the basal compartment under the insert membrane and assumes that substantial mixing occurs only during sample removal and fresh medium injection. The parameters providing the best fit for a transwell insert within a 12-well plate are summarized in Table 2. These parameters, together with the model Equations (Equation 8)–(Equation 14), can be used to estimate diffusive fluxes from the measured TB501 content of the samples (Figure 3c). The plot of the model-estimated transmembrane flux $I_{i}^{b}$ as a function of $\Delta c_{i}$ reveals a linear relationship between these quantities with a slope of 5.66 μL/min (Figure 3d). This value, normalized by the membrane area of A=0.6 cm${}^{2}$, translates to a diffusive permeability of $P_{0}=5.66\pm 0.28$ mm/h.

The model predicts a substantial amount of TB501 adsorbed to the transwell membrane. This prediction was tested by determining the amount of TB501 still in solution in the apical and basolateral compartments after 6 h of diffusion from the apical compartment. To measure the amount of TB501 adsorbed to the transwell device, we prepared three membranes by soaking them with CM for 5 days and kept another three membranes in ICM. After introducing the TB501 solution (10 μM) to the apical compartments, we kept the culture plate on a rotary shaker within a cell culture incubator. The transwell inserts were subjected to a 200 rpm agitation to improve mixing between the basal and lateral compartments. However, the movement was not vigorous enough to displace the culture insert within the culture well or to elevate the medium surface to an extent, allowing direct fluid exchange between the transwell compartments. After 6 h, we collected the medium from the apical and basolateral compartments and evaluated their TB501 content using absorption spectrophotometry. The results are summarized in Figure 3e and indicate that around 20% of the compound is adsorbed to the transwell device, this amount being slightly reduced when serum proteins were present during pre-incubation.

### 3.4. Transport through the VERO E6 Barrier

As a cellular barrier model system, we seeded n=3 transwell cultures with VERO E6 kidney tubule epithelial cells and monitored their maturation by measuring TEER values daily. When the TEER values reached a pre-determined threshold of 10 Ω cm${}^{2}$, 5 nmol TB501 was administered into the apical compartment, followed by the periodic sample collection process. The sub-compartment model was used to analyze the TB501 content of the samples (Figure 4a) and to extract the trans-barrier flux (Figure 4b) and the amounts of adsorbed compound (Figure 4c). As Figure 4a reveals, the presence of the cell layer delays the concentration peak in the basolateral compartment and reduces the overall permeability of the barrier. Surprisingly, the estimated trans-barrier current as a function of the concentration difference across the barrier is still linear but does not pass through the origin (Figure 4d). We interpret the negative y-intercept as a flux (0.5±0.1 nmol/h cm${}^{2}$) corresponding to the TB501 metabolized by the cells. The overall permeability coefficient of the VERO barrier is P=3±0.2 mm/h. Remarkably, when the model is used to fit the TB501 content of samples obtained from individual cultures, the model-predicted permeability coefficients are inversely related to the corresponding TEER value of the culture (Figure 4d, inset).

### 3.5. Mathematical Model of Diffusion through a Transwell Barrier Culture

A mathematical model of diffusive transport through a transwell barrier culture (Figure 5a,b) can characterize the cell layer in more detail than the overall permeability coefficient *P* does. Let us denote the concentration in the apical and basal compartments by $c_{a}$ and $c_{b}$, respectively. Similarly, let $c_{A}$ and $c_{B}$ denote the concentration within the apical and basal cell membranes, while c(h) denotes the average concentration at position *h* along the apical–basal axis. Using a simplified (far from saturation) Langmuir model of surface adsorption [44], the uptake rate at the apical membrane of the cell layer is $k_{on}^{cell}\cdot c_{a}$, while the desorption rate is $k_{off}^{cell}\cdot c_{A}$. The ratio of the rate parameters $K^{cell}=k_{on}^{cell}/k_{off}^{cell}$ characterizes the affinity of the binding. In equilibrium, the adsorption and desorption rates are equal:(18)$$k_{off}^{cell}\cdot c_{A}=k_{on}^{cell}\cdot c_{a},$$
hence,
(19)$$c_{A}=K^{cell}\cdot c_{a}.$$

If the transport of TB501 is passive, a diffusion equation
(20)$$\frac{\partial c}{\partial t}=D\cdot \frac{\partial^{2}c}{\partial h^{2}},$$
describes the time evolution of the concentration profile c(h,t), where *D* is the (spatially averaged) diffusivity of the TB501 within the barrier. The diffusion Equation (Equation 20) has mixed boundary conditions; the concentration is set at the apical side using Equation (Equation 19). The volume of the pores in the transwell membrane is so small that we neglect their reservoir capacity, hence, the diffusive current at the cellular basal surface equals the diffusive current across the membrane of the transwell insert:(21)$$J=-A\cdot D\cdot {\left.\frac{\partial c}{\partial x}\right|}_{h=h_{B}}=P_{0}\cdot A\cdot \left(\frac{c_{B}}{K^{cell}}-c_{b}\right),$$
where $P_{0}$ is the diffusive permeability of the transwell membrane (Figure 3d), *A* is the area and $H=h_{B}-h_{A}$ is the thickness of the cell layer (see Figure 5b). We assumed that the relation between the basal cell membrane concentration, $c_{B}$, and the concentration at the membrane pores is analogous to Equation (Equation 19). Finally, concentration changes in the apical and basal compartments are determined using the balance equations
(22)$$V_{b}\cdot \frac{dc_{b}}{dt}=J,$$
(23)$$V_{a}\cdot \frac{dc_{a}}{dt}=A\cdot D{\left.\frac{\partial c}{\partial h}\right|}_{h=h_{A}}-A\cdot M,$$
where $V_{a}$ and $V_{b}$ denote the volume of the well-mixed apical and basal liquid compartments, and *M* is the rate at which cells metabolize TB501 per unit area. The model also predicts the total amount of compound within the cell layer as
(24)$$N=A\int_{h_{b}}^{h_{a}}c\left(h\right)\cdot dh.$$

Finally, the diffusive permeability can be also obtained from the model using Equation (Equation 21) and the definition of *P* as
(25)$$J=PA(c_{a}-c_{b})$$
yielding
(26)$$P=P_{0}\frac{DK^{cell}}{HP_{0}+DK^{cell}}.$$

**Figure 5 pharmaceutics-15-02646-f005:**
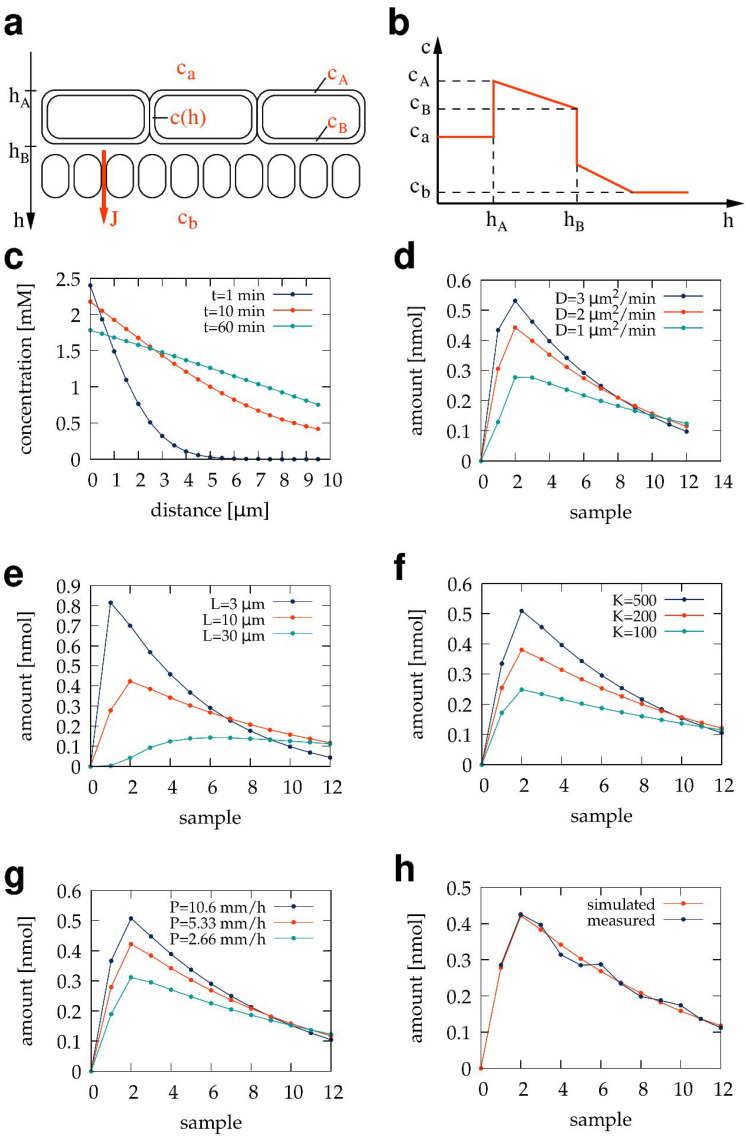
Mathematical model of a transwell barrier culture. (**a**,**b**) Schematic representation of the cell layer and the concentration profile c(h), the porous transwell membrane, and the diffusive transmembrane current *J*. (**c**) Model-predicted concentration profiles across a 10 μm thick cell layer 1, 10, and 60 min after introducing TB501 into the apical chamber. According to the simulations, a linear concentration profile develops within minutes. (**d**–**g**) Simulated transwell currents for various values of the diffusivity *D* (**d**), cell layer thickness *H* (**e**), affinity $K^{cell}$ (**f**), and membrane permeability $P_{0}$ (**g**). In each panel, only one parameter is changed according to the figure key. (**h**) Model simulations with parameter values in Table 3 (red) fit the estimated VERO-E6 transport current data (blue).

**Table 3 pharmaceutics-15-02646-t003:** Parameters fitted to characterize a 10 μm thick VERO E6 cell layer in a barrier culture.

Parameter	Value
$K^{cell}$	250
*D*	1.8 μm${}^{2}$/min
$P_{0}$	5.33 mm/h
*H*	10 μm
*M*	0.81 nmol/h cm${}^{2}$

For the given parameters *D*, *H*, $P_{0}$, $K^{cell}$, and *M*, the diffusive cell transport model can be solved numerically with the initial conditions $c_{b}=c_{B}=c\left(h\right)=0$, and $c_{a}=10$ μM. Figure 5b demonstrates the concentration profiles developing in the simulations using the parameter values in Table 3. For these parameters, a linear concentration profile develops in the cell layer within minutes. After this initial transient, the slope of the linear concentration profile slowly changes with time, reflecting the corresponding changes in $c_{a}$ and $c_{b}$.

To adapt the above model to our experimental setup, at a pre-determined sequence of sampling timepoints $t_{i}$, we store the TB501 inflow as
(27)$$J_{i}=c_{b}\left(t_{i}\right)V_{b}$$
and then set the basal concentration to zero:(28)$$c_{b}\left(t_{i}\right)=0.$$

Model simulations with various parameter values are presented in Figure 5c–f. Increasing the value of *D*, $P_{0}$, and $K^{cell}$, or decreasing *H*, all increase the initial peak of the diffusive current. The subsequent decrease in the current *J* reflects the gradual loss of TB501 from the apical compartment.

Model parameters can be estimated (Figure 5g) by fitting the model-predicted $J_{i}$ sequence to the estimated transwell currents $I_{i}^{b}$ in experiments by minimizing the total error of the fit $\Delta (J,I^{b})$, as well as the difference between the extrapolated infinite simulated sample sequence and the amount of reagent that was initially in the apical compartment $\Delta_{a}\left(J,V_{a}c_{a}\left(t=0\right)\right)$. Some parameters have analogous effects; the effect of increasing diffusivity *D* is very similar to the effect of decreasing the cell layer thickness *H*. Therefore, the fitting procedure could not determine all five model parameters. Relying on histological data, however, allows us to fix the value of H=10 μm, and that makes fitting the rest of the parameters possible. Parameter values yielding the best fit for the VERO E6 barrier are listed in Table 3.

## 4. Discussion

### 4.1. Permeability Coefficient and Kinetic Parameters

The transport of a compound through a diffusion cell is often characterized by the permeability coefficient
(29)$$P=\frac{Q(T)}{AT\Delta c}$$
where Q(T) is the amount of compound collected during a time interval of length *T* in the basolateral or acceptor compartment, *A* is the area of the sample, and Δc is the concentration difference of the compound between the two surfaces of the barrier sample [9,10,14]. This approach works best when Δc is relatively steady during the duration *T* of the measurement, which can be achieved using suitably large apical (donor) and basal (acceptor) compartments, as in the case of Franz chambers. However, this choice also requires using large amounts of test compounds and employing sensitive detection methods to measure the concentration of the highly diluted compound in the basal (acceptor) compartment.

When utilizing miniaturized diffusion chambers, changes in Δc can be substantial. Hence, the accumulation of the compound, Q(T), is not linear in time, and the estimate Equation (Equation 29) depends on the particular choice of *T* [14,15]. The sub-compartment model and the analysis it enables (Figure 3d and Figure 4d) estimate the permeability *P* and the metabolic degradation of the test compound *M* using time-resolved concentration data.

Depending on the time resolution of the sampling procedure, Q(T) also deviates from the behavior described by Equation (Equation 29) for small sampling times *T* as the diffusion process across the barrier requires some time to reach a steady state (Figure 4b and Figure 5b). These deviations can be utilized with the help of the mathematical model depicted in Figure 5a to estimate the compound diffusivity *D* within, and the affinity $K^{cell}$ to, the cells of the barrier layer.

Our estimated permeability coefficient for the TB501 through VERO E6 cells, P=3 mm/h, is similar to the reported value for a small molecular weight organic compound (Methyl 4-hydroxybenzoate, $M_{w}=152.1$ Da) through a skin sample [9]. Kidney cell lines were reported to exhibit permeability coefficients in the range of 0.03 mm/h to 1 mm/h when assayed with mannitol, a non-cell-permeable marker compound [45]. Thus, the permeability coefficient obtained in our pilot study is of a similar magnitude to those reported previously for similar small molecular weight compounds in barrier cultures and substantially higher than those reported for non-cell-permeable compounds.

### 4.2. Comparison of Transwell Inserts and Franz Chambers

While both transwell inserts and Franz chambers are devices used to study biological barriers, their size, culture conditions, and mechanical requirements for the sample studied segregated their usage into distinct fields within pharmacology. Here, we demonstrated that, with proper sampling and mathematical corrections, regular transwell cell cultures can provide the information usually extracted from Franz chamber measurements [14,15,16,17,18,19,20,34,35,36]. Our estimate that 95% of the basal volume is removed during the automated sampling procedure ($q^{b}=5.3\%$, Table 2) indicates that the basolateral compartment can be considered well-mixed on a time scale exceeding the length of the sampling period.

### 4.3. Adsorption of the Test Compound

Our analysis also indicated the importance of taking into account the adsorption of the test compound to the surfaces of cell culture and fluidics devices ([4], Figure 3c,e and Figure 4c). Any fluidics system with organic polymer tubing can adsorb apolar compounds, but in our case, the most important adsorbent surface is the culture insert itself. Our results are consistent with previous reports analyzing the adsorption of apolar compounds to membranes of sterile filters [46].

While the mathematical models schematically presented in Figure 2a,b greatly simplify the complexities of a three-dimensional fluid flow, their applicability is supported by consistently incorporating several measurable aspects of the system as follows. (1) The same parameters describe compound dilution experiments irrespective of the initial compound concentration at the onset of the experiments (Figure 2b). (2) The ability to estimate the external input from the compound content of the basolateral samples (Figure 2d), and (3) to describe experiments where the compound was initially added either to the basolateral (Figure 3b) or to the apical (Figure 3c) compartments. (4) The calculated diffusive transmembrane flux of the compound is proportional to the transmembrane concentration difference (Figure 4d). (5) Correctly estimating that a substantial portion of the compound is adsorbed to the transwell insert (Figure 4e).

The specific form of the model equations does not assume that the medium is in equilibrium with the adsorbent surfaces. In fact, Figure 3c and Figure 4c predict that the transwell device continues to adsorb compounds for several hours. Similarly, the interaction between the fluid sample and the tube walls of the fluidic system is short enough that only a portion of the adsorbed amount can re-enter the solution (Table 1). Rinse experiments with a ten times higher initial concentration of the test compound exhibited the same behavior (Figure 2b), suggesting that the saturation of the adsorbing surface is not a major factor in our experimental setup.

### 4.4. Diffusion through a Barrier Culture

Models similar to the one depicted in Figure 5 were proposed previously [47,48]. Here, we intended to keep the model as simple as possible to be able to fit its parameters to the experimental data. Despite the model’s simplicity, we could only unambiguously determine some parameters after we fixed the cell layer thickness *H*.

Independent considerations can verify some of the values established in Table 3. The fitted metabolic rate, M=0.81 nmol/h·cm${}^{2}$, is within 50% of the value (0.5 nmol/h cm${}^{2}$) extracted from the ordinate intercept in Figure 4d. Furthermore, the fitted values in Table 3 can be combined into the diffusive permeability using Equation (Equation 26), yielding P≈2 mm/h, which is, again, within 50% of the value extracted from the slope of Figure 4d. One potential source of this discrepancy is the significant amount of test compound accumulating within the barrier layer, *N*. While the diffusion model (Figure 5a) provides an estimate in Equation (Equation 24), the sampling model presented does not consider it in Equations (16) and (17). Future work can develop a fitting procedure that handles both the diffusive model of the barrier layer and the adsorption effects of the sampling process.

## 5. Conclusions

In this pilot study, we employed an automated millifluidic system to characterize the transport through a VERO E6 kidney model barrier culture. The test compound was an antimycobacterial drug candidate, TB501. This approach could obtain enough samples from regular transwell barrier cultures to determine their time-resolved transport kinetics. We also developed a mathematical model of the sample collection process that can be used to estimate and compensate for adsorption artifacts and thus provide a more accurate estimate of the diffusive transport. The time-resolved concentration data allowed us to determine parameters of the cellular barrier, such as the diffusive permeability of the test compound, its affinity to the cell membrane, its diffusivity through the cellular barrier, and the rate of its metabolic elimination. We expect the presented method to be fruitful in evaluating other compounds with different chemical features using simple barrier cultures. The proposed mathematical model can also be extended to include various forms of active transport.

## Figures and Tables

**Figure 1 pharmaceutics-15-02646-f001:**
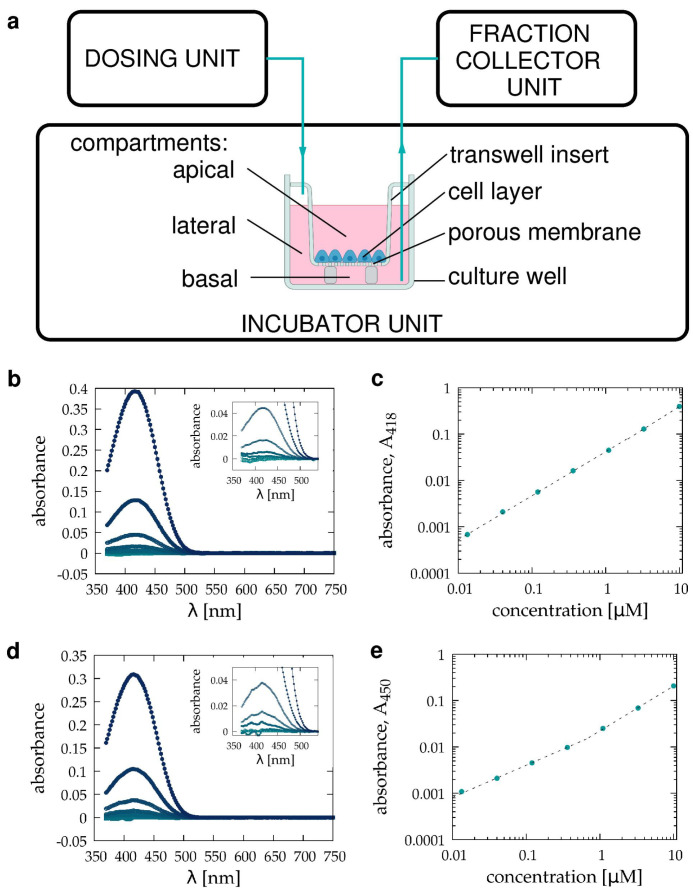
Experimental setup to characterize the transport of TB501, a drug candidate, through a barrier model of VERO E6 cells. (**a**): The automated sample collection utilizes a dosing and a fraction collector unit outside the cell culture incubator. A separate tubing network injects fresh medium into and removes the TB501-containing medium from the basolateral compartment of a transwell chamber. For mathematical analysis, we distinguished the basal and lateral compartments of the basolateral volume. (**b**–**d**): TB501 absorbance spectra, referenced to the appropriate medium. The compound was diluted to 13 nM, 40 nM, 120 nM, 360 nM, 1.1 μM, 3.3 μM, and 10 μM concentrations, indicated by light to dark blue colors, in ICM (**b**,**c**) and HPMI (**d**,**e**). Insets show a portion of the spectra magnified near the 418 nm peak. Absorbance values at 418 and 450 nm can be used to determine TB501 concentration (**c**,**e**). Dashed segmented lines indicate linear and power law fits for the strong and weak absorbance regimes, respectively.

**Figure 4 pharmaceutics-15-02646-f004:**
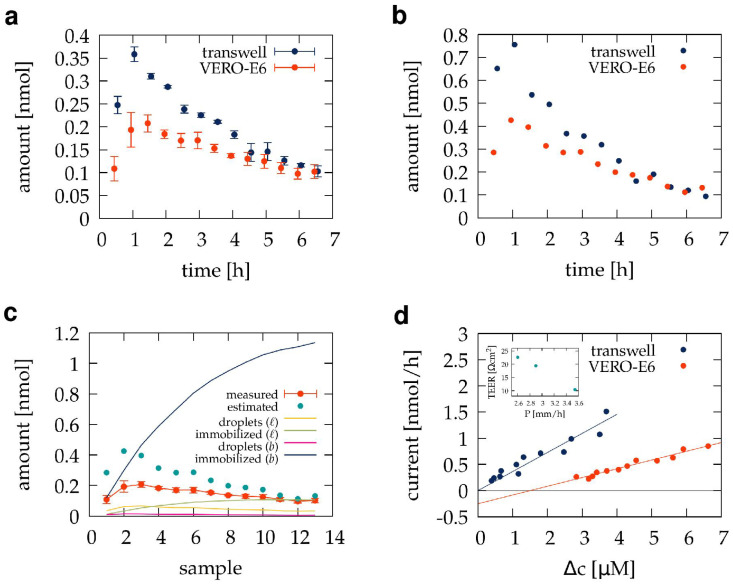
VERO E6 cultures as diffusion barriers. (**a**) The amount of TB501 in basolateral samples obtained from pre-conditioned transwell inserts (blue) and VERO-E5 cultures (red). Each data point is the average of n=3 independent experiments. (**b**) Transmembrane TB501 fluxes over the 30 min sampling period, estimated by fitting the mathematical model of Figure 3 to the average TB501 content of sample sequences shown in panel (**a**). (**c**) Localization of TB501 in the VERO E6 barrier model. Solid lines represent mathematical model predictions for the TB501 content of the samples (red), the amount of TB501 adsorbed to the transwell membrane (dark blue), to the culture and fluid handling system (yellow), or remaining in droplets either under the transwell membrane (pink) or in the rest of the system (green). (**d**) Model-predicted diffusion permeability of the VERO E6 barrier culture (red), compared with that of a pre-conditioned transwell membrane (blue). The calculated permeability coefficients (*P*) and TEER values of individual cultures exhibit an inverse relation (inset).

## Data Availability

All relevant data are included in the manuscript. Raw measurement data and computational scripts are available at the Open Science Framework (https://osf.io/sc3wq, accessed on 11 November 2023).

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
