# Peer review of "Quantitative Analysis of a Pilot Transwell Barrier Model with Automated Sampling and Mathematical Modeling"

_pharmaceutics, 2023, doi:10.3390/pharmaceutics15112646_

Round 1
Reviewer 1 Report (Previous Reviewer 1)
Comments and Suggestions for Authors
This is a detailed interrogation of an interesting topic
The paper is detailed and I have only a couple of minor revisions
1. At times the methodology is repeated within the results section eg section 3.1 This can make the paper hard to read and a little repetitive
2. Page 16 line 308-311 the permeablity is compared to being like through skin whereas the cell line is not a skin cell line - was this expected?
3. The generalisability of hte method could be more explicit in terms of how this could be used for other cell lines and drugs
4. the assumption of passive permeability is a big one and needs to be stated even within the abstract
Author Response
1. At times the methodology is repeated within the results section eg section 3.1 This can make the paper hard to read and a little repetitive
We moved some of the technical details from Sec 3.1 and merged them with the text in Section 2.2 in Materials and Methods. Some repetition is unavoidable as the Results and Figures should have an overall description of the experiments reported.
2. Page 16 line 308-311 the permeablity is compared to being like through skin whereas the cell line is not a skin cell line - was this expected?
We now also compare the permeability values with data reported for mannitol, a non-cell-permeable marker compound, in various kidney barrier cultures. We conclude that "the permeability coefficient obtained in our pilot study is of a similar magnitude reported previously for similar small molecular weight compounds in barrier cultures and substantially higher than reported for non-cell-permeable compounds." We believe, however, that the quality of the barrier culture model affects the permeability values more than the nature of the epithelium it represents; thus, a leaky skin barrier model can yield permeability values similar to a tight-closing kidney barrier model.
3. The generalisability of hte method could be more explicit in terms of how this could be used for other cell lines and drugs
We agree and state this both in the abstract and the conclusions: "We expect that the presented method will be fruitful in evaluating other compounds with different chemical features using simple barrier cultures. The proposed mathematical model can also be extended to include various forms of active transport."
4. the assumption of passive permeability is a big one and needs to be stated even within the abstract.
Accordingly, the rephrased version reads: "The large number of samples collected allows us to fit a detailed mathematical model of the passive diffusive currents to the measured concentration profiles." Furthermore, the last sentence emphasizes that "The proposed mathematical model can also be extended to include various forms of active transport."
Reviewer 2 Report (Previous Reviewer 3)
Comments and Suggestions for Authors
The content of the revised version has been improved. Please enhance some paragraphs.
1. Figure 1(e), it is not a curve. There are two linear lines, please replot them.
2. “Table 1: Fitted parameters characterizing the sample collection process, both in the absence andpresence of a transwell insert.”, Please provide more detailed information on how to find these parameters. Same as Table 2.
3. Try to edit equations (16) and (29) to section 2. Materials and methods
Comments on the Quality of English LanguageMinor editing of English language required
Author Response
1. Figure 1(e), it is not a curve. There are two linear lines, please replot them.
We modified Figs 1c and 1e to represent our fits as a segmented line. We modified the figure legend accordingly.
2. “Table 1: Fitted parameters characterizing the sample collection process, both in the absence and presence of a transwell insert.”, Please provide more detailed information on how to find these parameters. Same as Table 2.
The fitting procedure is described in detail in Section 2.6. Furthermore, our raw data and the code used for analysis are shared in the Open Science Framework as stated in the paper.
3. Try to edit equations (16) and (29) to section 2. Materials and methods
We moved Eqs (16) and (17) into the Methods section 2.6.2. Equation (29), however, is meaningful only after the definition of our mathematical model in Section 3.5, where variables D and K are introduced -- so we moved it to the end of Section 3.5.
This manuscript is a resubmission of an earlier submission. The following is a list of the peer review reports and author responses from that submission.
Round 1
Reviewer 1 Report
Comments and Suggestions for Authors
This is an interesting paper but I suggest the following revisions
The abstract and introduction need to include details on the barrier that you aim to describe. My first impression was that this would be an oral permeability assay (comparable to Caco-2 monolayer) but the introduction describes skin as the barrier then the cells are of kidney origin. Please be explicit in which barrier you are replicating in your system. This needs to be explicit throughout the paper
The pH of the transport media need to be reported as this can influence the solubility and hence transport of drugs
Section 3.4 did you consider that hte drug may be trapped within the cells which may contribute to the delay and hte absoprtion previously reported?
In the discussion you revert back to skin samples but I am unsure as to whether this was as a function of the permeability observed or the structure of your cell model?
Section 4.2 in comparison to Franz chambers there is no data - it would be great to compare your system to Franz cell data if this is possible using data in the literature
Your model will only be useful once it has been compared to existing systems and has some correlation to in vivo data - the current manuscript only contains pilot data.
Selection of a model drug that does not adhere to the plasticware would really help in the robustness of your model and interpretation of data
Comments on the Quality of English Language
Some minor edits required to make it explicit the hypothesis under test
Author Response
* The abstract and introduction need to include details on the barrier that you aim to describe. My first impression was that this would be an oral permeability assay (comparable to Caco-2 monolayer) but the introduction describes skin as the barrier then the cells are of kidney origin. Please be explicit in which barrier you are replicating in your system. This needs to be explicit throughout the paper
We substantially rewrote both the abstract and the introduction, and included this information in the title as well. Now, we state in the abstract that: "In this pilot study, the transport properties of TB501, an antimycobacterial drug candidate, were characterized using an in vitro barrier model of VERO E6 kidney cells. "
In the introduction, we now explain that "Currently, the two main experimental approaches to analyze trans-barrier transport utilize either cell cultures and transwell diffusion chambers or excised tissue pieces placed into much larger devices like the Franz and Ussing chambers." We only refer to Franz chamber studies as the kinetic parameters we determine are typically obtained from such systems.
* The pH of the transport media need to be reported as this can influence the solubility and hence transport of drugs
We now include our media's pH (7.4) in the Materials and Methods section.
* Section 3.4 did you consider that hte drug may be trapped within the cells which may contribute to the delay and hte absoprtion previously reported?
We appreciate this comment and included a formula in Eq. (24) for calculating this amount in our diffusive barrier layer model. When we calculate the transmembrane fluxes using the sub-compartment model the amount taken up by the cells is not considered in Eqs. (16) and (17) which may also affect Fig 4d. As we now discuss, this may explain the discrepancy in the estimates for the metabolic rate and diffusive permeability in Sections 2.4 and 2.5. Ideally, a fitting procedure could be developed that takes into account both the adsorption effects during sample collection and the detailed diffusive model through the cell layer. However, this endeavor is outside of the scope of the current paper.
* In the discussion you revert back to skin samples but I am unsure as to whether this was as a function of the permeability observed or the structure of your cell model?
We revert to studies where similar, time-resolved kinetic data are collected. These just happen to be skin samples in Franz chambers. We consider our method equally applicable to a wide variety of biological barriers, not just the kidney cell layer employed in this pilot study.
* Section 4.2 in comparison to Franz chambers there is no data - it would be great to compare your system to Franz cell data if this is possible using data in the literature
We now include references where similar quantities were extracted -- but employing a different barrier and agents.
* Your model will only be useful once it has been compared to existing systems and has some correlation to in vivo data - the current manuscript only contains pilot data.
We agree -- and modified the title and our wording accordingly.
* Selection of a model drug that does not adhere to the plasticware would really help in the robustness of your model and interpretation of data
We agree with this comment: We plan to conduct future studies that will expand the usability of the proposed method beyond the current pilot state.
Reviewer 2 Report
Comments and Suggestions for Authors
The manuscript presents a method of measurement of transwell transport with automated sampling and mathematical description and modeling. The method and model can be useful for researchers studying transport through physiological barriers.
Remarks:
What apparatus was used for transport measurements? Only the inserts are described.
The medium in the basolateral compartment was replaced every 30 minutes. This information is given only in the abstract. It should be given under Material and methods.
Line 12: please change “comound” to “compound”
Line 65: please change “obtained” to “obtained”
Line 78: “9mM”, please separate
Formulas (1), (2): please explain “min”; what does “,” mean in the formulas?
Line 130: “2ml”, please separate
Comments on the Quality of English LanguageI detected some spelling errors; so it is better to check the text carefully
Author Response
* What apparatus was used for transport measurements? Only the inserts are described.
We better specified the device in the Materials and Methods section "Automated sample handling".
* The medium in the basolateral compartment was replaced every 30 minutes. This information is given only in the abstract. It should be given under Material and methods.
In the resubmitted version, this information is included in the Materials and Methods section "Transwell culture" and the first "Pilot transwell transport measurement" section of the Results.
* Line 12: please change "comound" to "compound"
* Line 65: please change "obtained" to "obtained"
* Line 78: “9mM”, please separate
* Line 130: “2ml”, please separate
These typos are corrected in the resubmitted version.
* Formulas (1), (2): please explain "min"; what does "," mean in the formulas?
We expanded the Materials and Methods section "Spectroscopy":
"The absorbance calibration data can be fitted with a linear function for absorbance values above 0.03 and with a power law below (Fig. 1). Accordingly, we used either the linear or the power law fit to calculate concentrations depending on the magnitude of absorbance. "And also defined the min function used in Eqs (1) and (2).
Reviewer 3 Report
Comments and Suggestions for Authors
The content of this paper was to report the quantitative analysis of a transwell barrier model with automated sampling. It provides novel theories and useful information. However, it was different to read and understand at this stage. Please enhance the content with writing skills.
1.The title could be changed to “Quantitative analysis of a transwell barrier model with automated sampling and modeling.”
2. The quality of Figure 2,3,4,5 need to be improved.
3. All equations 3-28 should be moved to the section of Materials and Methods as subsection as “Theoretical analysis” or “theoretical background”.
4. Section 5 Conclusion must be written in the style of a journal.
5. Please check the grammar and style of English very carefully.
Comments on the Quality of English LanguageExtensive editing of English language required. Many mistakes were found.
Author Response
1.The title could be changed to "Quantitative analysis of a transwell barrier model with automated sampling and modeling."
We changed the title to "Quantitative analysis of a kidney transwell barrier model with automated sampling and mathematical modeling." We feel it is important to emphasize that the models are mathematical -- the transwell barrier is also a model of a biological barrier layer.
2. The quality of Figure 2,3,4,5 need to be improved.
The submitted pdf document contained only low-resolution versions of the figures. We now supply them in full resolution -- but also increased the size of most symbols on the plots.
3. All equations 3-28 should be moved to the section of Materials and Methods as subsection as "Theoretical analysis" or "theoretical background".
We moved most of the equations to two new sections in the Materials and Methods, which certainly improved the readability of the Results section. However, we kept the diffusive model in the Results, as it is indeed a new and important result considering the focus of the manuscript.
4. Section 5 Conclusion must be written in the style of a journal.
We have expanded the Conclusions accordingly.
5. Please check the grammar and style of English very carefully.
We have corrected several grammatical errors and checked the manuscript extensively.